# Violation of emergent rotational symmetry in the hexagonal Kagome superconductor $CsV_3Sb_5$

Kazumi Fukushima[1], Keito Obata[1], Soichiro Yamane[1,2], Yajian Hu[1], Yongkai Li[3,4,5], Yugui Yao [3,4], Zhiwei Wang [3,4,5] ✉, Yoshiteru Maeno[1,6] & Shingo Yonezawa [1,2] ✉

Superconductivity is caused by electron pairs that are canonically isotropic, whereas some exotic superconductors are known to exhibit non-trivial anisotropy stemming from unconventional pairings. However, superconductors with hexagonal symmetry, the highest rotational symmetry allowed in crystals, exceptionally have strong constraint that is called emergent rotational symmetry (ERS): anisotropic properties should be very weak especially near the critical temperature $T_c$ even for unconventional pairings such as $d$-wave states. Here, we investigate superconducting anisotropy of the recently-found hexagonal Kagome superconductor $CsV_3Sb_5$, which is known to exhibit various intriguing phenomena originating from its undistorted Kagome lattice formed by vanadium atoms. Based on calorimetry performed under accurate two-axis field-direction control, we discover a combination of six- and two-fold anisotropies in the in-plane upper critical field. Both anisotropies, robust up to very close to $T_c$, are beyond predictions of standard theories. We infer that this clear ERS violation with nematicity is best explained by multi-component nematic superconducting order parameter in $CsV_3Sb_5$ intertwined with symmetry breakings caused by the underlying charge-density-wave order.

## Anisotropies in superconductors

Superconductivity occurs as a consequence of the formation of electron pairs called the Cooper pairs. In standard microscopic theories, Cooper pairs are assumed to be isotropic[1]. Similarly, standard theories for macro- or mesoscopic properties, the Ginzburg-Landau (GL) formalisms, also canonically assume isotropic macroscopic quantum-mechanical wavefunction as the superconducting (SC) order parameter[2]. Experimentally, indeed, most ordinary superconductors exhibit rather isotropic SC properties or, at most, trivial anisotropy

inherited from normal-state electronic properties[3]. However, some exotic superconductors exhibit non-trivial anisotropic properties stemming from order-parameter structures, and both microscopic as well as phenomenological theories have been extended to deal with such unconventional superconductivity. A well-known example is the $d$-wave SC state having sign changes in the order parameter upon 90-degree rotation, hosting various interesting anisotropic properties[4,5]. Another exotic example is nematic superconductivity[6], which is recently identified first in doped $Bi_2Se_3$ superconductors[7–10] and

[1]Department of Physics, Graduate School of Science, Kyoto University, Kyoto 606-8502, Japan. [2]Department of Electronic Science and Engineering, Graduate School of Engineering, Kyoto University, Kyoto 615-8510, Japan. [3]Key Laboratory of Advanced Optoelectronic Quantum Architecture and Measurement, Ministry of Education (MOE), School of Physics, Beijing Institute of Technology, Beijing 100081, P. R. China. [4]Beijing Key Lab of Nanophotonics and Ultrafine Optoelectronic Systems, Beijing Institute of Technology, Beijing 100081, P. R. China. [5]Material Science Center, Yangtze Delta Region Academy, Beijing Institute of Technology, Jiaxing 314011, P. R. China. [6]Toyota Riken-Kyoto University Research Center (TRiKUC), Kyoto University, Kyoto 606-8501, Japan. ✉e-mail: zhiweiwang@bit.edu.cn; yonezawa.shingo.3m@kyoto-u.ac.jp

subsequently in other systems such as bilayer graphene[11]. Nematic superconductors exhibit twofold anisotropy in bulk SC properties originating from rotational symmetry breaking in the order-parameter amplitude. Finding a new species of superconductivity with unique anisotropy is one of the most important goals of fundamental research on superconductivity.

Hexagonal materials are exceptionally unique in the study of novel superconductivity. On the one hand, the sixfold rotational symmetry, the highest rotational symmetry allowed in crystals, enables the system to realize interesting SC order parameters. In particular, various two-component superconducting order parameters, such as chiral or nematic $p$, $d$, or $f$-wave superconductivity, are allowed owing to the degeneracy between $(x, y)$ or $(xy, x^2 - y^2)$ basis functions[12]. On the other hand, hexagonal systems have a hidden special feature: sixfold rotational symmetry is equivalent to cylindrical symmetry up to the fourth-order terms in the GL theory[13–17]. Thus, for example, the upper critical field $H_{c2}$ in the plane perpendicular to the sixfold axis should be perfectly isotropic. This isotropy can be removed by considering sixth-order terms, allowing the presence of weak sixfold $H_{c2}$ anisotropy. Nevertheless, such in-plane hexagonal $H_{c2}$ anisotropy, $H_{c2}^{(6)}$, if exists, should rapidly vanish for $T \to T_c$ as $H_{c2}^{(6)} \propto (1 - T/T_c)^3$ [15,16]. This phenomenon is called the emergent rotational symmetry (ESR) in hexagonal superconductors[17]. ESR is satisfied even if multiple Fermi surfaces exist or the Fermi velocity has anisotropy on the Fermi surfaces[18]. ESR violation, namely sixfold anisotropy that is robust with increasing temperature, is fully non-trivial. Experimentally, in-plane anisotropy in various hexagonal superconductors have been tested[19–24], but hexagonal anisotropy has only been reported in a very few examples using transport measurements[19,20,23]; and it has never been reported using thermodynamic probes. Moreover, the detailed temperature dependence of hexagonal anisotropy has never been elucidated.

Here, in this Article, we report the discovery of combined hexagonal and nematic SC properties in the hexagonal material CsV$_3$Sb$_5$. Our specific heat measurements performed under precise two-axis control of magnetic field directions reveal six- and twofold anisotropies in bulk $H_{c2}$ when the field is rotated in the hexagonal $ab$ plane. Our results marks the first thermodynamic evidence for the violation of the ERS in hexagonal superconductors, indicating the realization of SC states beyond the ordinary GL formalism. We argue that our result

is best explained by a two-component order parameter coupled to the underlying charge-density-wave (CDW) order, which is believed to weakly break the nominal hexagonal symmetry.

## Kagome superconductor CsV$_3$Sb$_5$

Our target material CsV$_3$Sb$_5$ (Figs. 1a, b), along with its sister compounds KV$_3$Sb$_5$ and RbV$_3$Sb$_5$, have been extensively studied recently because of their fascinating properties originating from undistorted Kagome net of vanadium (Fig. 1c), unconventional CDW order, and superconductivity[25,26]. In CsV$_3$Sb$_5$, the CDW occurs below 94 K, followed by superconductivity below $T_c \sim 3$ K. Both the CDW and SC states are of utmost interest, and the symmetry properties in these orders have been widely debated[27]. The CDW is now believed to be a bond order in the kagome lattice, resulting in the so-called star-of-David (SoD) or tri-hexagon (TrH) deformation[28,29]. The CDW also has four-unit-cell $c$-axis modulation, and thus it is characterized by a $2 \times 2 \times 4$ expansion of the unit cell[28]. The detailed crystal structure in the CDW phase is still under debate. Recent studies suggest either a trigonal model with the $P\bar{3}$ space group[28] or an orthorhombic model with the $Cmmm$ space group[30]. Additional time-reversal-symmetry breaking[31] and nematicity (rotational symmetry breaking) has also been reported[32–34]. Nevertheless, even in the CDW phase, the overall lattice distortion, as well as accompanying changes in the electronic state, is rather subtle[28,30,35,36], and the hexagonal symmetry remains a good starting point.

Reflecting the underlying symmetry breakings in this compound, the SC state can have a highly unconventional nature. However, the fundamental information, such as gap structure and/or order-parameter symmetry has not been established. Several reports claim fully-gapped conventional SC properties but with strong multi-band nature[37–39], whereas nodal properties have been suggested from thermal transport and scanning tunneling microscopy studies[40–42]. Moreover, the SC order parameter is recently found to accompany atomic-scale modulation, which is called the roton pair-density-wave (PDW) state[42]. To reveal the true nature of the superconductivity, the study of in-plane anisotropy is inevitably important because it reflects various properties of the SC order parameter[43]. For CsV$_3$Sb$_5$, several transport studies have reported twofold in-plane anisotropy in SC properties[32,44], suggesting nematic feature in the SC state. However, for quasi-two-dimensional (Q2D) materials such as CsV$_3$Sb$_5$, very accurate field

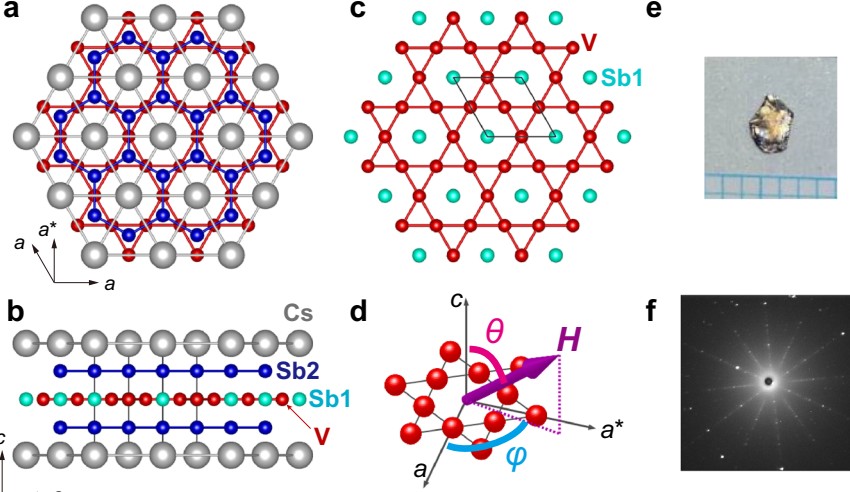

**Fig. 1 | Hexagonal crystal structure of the kagome superconductor CsV$_3$Sb$_5$.** **a**, **b** Crystal structure viewed from the $c$ axis (**a**) and from the $a^*$ axis (**b**). Cs and V are represented by the gray and red spheres. Sb are shown with light-blue (Sb1 site) and blue (Sb2 site) spheres. **c** Schematic of the vanadium kagome layer. The Kagome net of V is formed with Sb1 located at the center of hexagons. **d** Definition of the magnetic field angles $\theta$ and $\phi$. **e** Photo of the sample. The size of this sample is $3 \times 2 \times 0.2$ mm$^3$. The mass is 3.8921 mg. **f** Laue photo of the sample. Clear Laue spots indicate the high crystallinity of the sample.

alignment using two-axis field control is necessary to avoid any extrinsic features originating from field misalignment. To our knowledge, in-plane anisotropy studies performed under two-axis field-direction control have not been reported to date.

## Results

### Calorimetry using a high-quality sample

In this study, we investigated in-plane and out-of-plane anisotropies of the superconductivity in $CsV_3Sb_5$ by means of field-angle-resolved calorimetry. Specific heat $C$ was measured with a hand-made calorimeter based on the AC method. We used a high-quality single crystal grown by a self-flux method (Fig. 1e). Laue photos of this sample shows clear spots, as shown in Fig. 1f. We found that some other samples exhibits much broader spots, presumably due to stacking faults. The clear Laue spots in the present sample evidences high crystalline quality. Moreover, we have noticed that samples with broader Laue spots tend to exhibit broader CDW or SC transitions, indicating the importance of choosing samples with high crystallinity.

Figure 2a shows the magnetic susceptibility of this sample, and Fig. 2b the temperature $T$ dependence of the electronic specific heat $C_e$ divided by $T$. For the specific heat data, its raw data before subtraction of the phononic part are presented in Supplementary Fig. 1. Both susceptibility and specific heat exhibit a very sharp superconducting transition with $T_c = 2.8\,K$. The susceptibility reaches the value for the full Meissner screening below around 2.5 K. Similarly, $C_e/T$ extrapolates almost to zero as $T \to 0$, suggesting nearly 100% volume fraction. The sharp transition and almost perfect volume fraction again indicate high sample quality.

The observed specific heat data are compared with a prediction of the standard Bardeen–Cooper–Schrieffer (BCS) theory. We found a noticeable difference from the prediction of the BCS theory (broken curve in Fig. 2b): The observed jump $\Delta C_e/T_c$ divided by the normal-state electronic specific heat coefficient $\gamma$ is 1.0, smaller than the BCS prediction ($\Delta C_e/\gamma T_c = 1.43$). Moreover, $C_e/T$ below $T_c$ is rather linear in $T$ in contrast to the exponential decay in the prediction. These features contain information on the SC gap structure, which will be discussed elsewhere.

### Quasi-two-dimensional superconductivity

To study SC anisotropy, we used a vector magnet system with a horizontal rotation stage[45]. This system, the same one as used in ref. 8, allows us to perform accurate two-axis field-direction control. The field is in situ aligned to the sample by making use of the $H_{c2}$ anisotropy, and the accuracy of the field alignment is better than 0.1 degrees (Supplementary Figs. 2, 3). Throughout this paper, we define $\theta$ as the polar angle of the field measured from the $c$-axis and $\phi$ as the azimuth angle of the field along the $ab$ plane measured from the $a$-axis, as illustrated in Fig. 1d.

To evaluate the upper critical field $H_{c2}$, we measured the field strength dependence of $C/T$ under various field directions (exemplified in Supplementary Fig. 4). We then deduced $H_{c2}$ from the anomaly in the $C(H)$ curves. The obtained $H_{c2}$ for three different crystalline directions, namely the $a$, $a^*$, and $c$ axes, are plotted in Fig. 2c. As one can see, $H_{c2}$ for the in-plane $a$ and $a^*$ directions are quite similar but are nearly ten times larger than that for the $c$ direction. For more quantitative comparison, we plot the out-of-plane anisotropy parameter $\Gamma \equiv H_{c2\perp c}/H_{c2\parallel c}$ in the inset of Fig. 2c. The anisotropy amounts to $\Gamma = 11.4 \pm 0.2$ above 1.6 K and reduces weakly to 8.9 at the lowest temperature. Such temperature variation in $\Gamma$ is comparable to those observed in various layered superconductors[46].

The out-of-plane anisotropy is studied in more detail from the polar angle $\theta$ dependence of $H_{c2}$ shown in Fig. 2d. The $H_{c2}$-$\theta$ curve

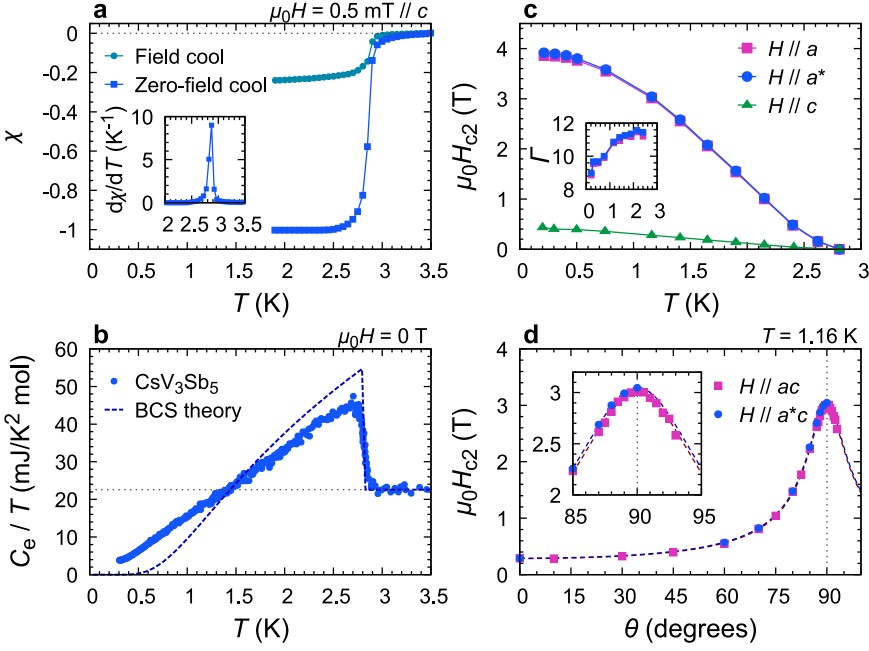

**Fig. 2 | Zero-field superconducting properties and out-of-plane anisotropy of $CsV_3Sb_5$. a** DC magnetic susceptibility $\chi$ of our $CsV_3Sb_5$ single crystal measured under zero-field-cooled and field-cooled conditions. The demagnetization factor of 0.88 estimated from the sample shape has been considered. The inset shows the temperature derivative of zero-field-cooled $\chi$, showing a sharp superconductive transition at the critical temperature $T_c = 2.8\,K$ with a full-width of half-maximum of less than 0.1 K. **b** Zero-field electronic specific heat of $CsV_3Sb_5$. The sharp jump at $T_c$ again indicates a high quality of the sample. The temperature dependence is clearly different from the prediction of the standard BCS theory (dotted curve)[3]. **c** Upper

critical field $H_{c2}$ along the $a$ (pink squares), $a^*$ (blue circles), and $c$ axes (green triangles). The inset shows the out-of-plane anisotropy $\Gamma \equiv H_{c2\parallel a}/H_{c2\parallel c}$ (pink squares) and $H_{c2\parallel a^*}/H_{c2\parallel c}$ (blue circles). This quantity ranges from 9 to 11.5, indicative of the Q2D nature of superconductivity. **d** Polar field-angle $\theta$ dependence of $H_{c2}$ at 1.16 K in the $ac$ (pink squares) and $a^*c$ planes (blue circles). Both data sets are well fitted by using an anisotropic mass model[3] indicated by the dotted curves. The fit yields the anisotropy parameter of 10.4 for the $ac$ plane and 10.5 for the $a^*c$ plane. The inset is an enlarged view near $\theta = 90°$.

measured at 1.16 K exhibits a sharp peak centered at $\theta = 90°$. The data were well fitted by the standard anisotropic mass model of the GL formalism: $H_{c2}(\theta) = H_{c2}(0°)/[\cos^2\theta + \Gamma^{-2}\sin^2\theta]^{1/2}$ (dotted curves in the figure). The fit yields $\Gamma = 10.5$, in agreement with the $\Gamma$ values in the inset of Fig. 2b. This successful fitting of the anisotropic mass model indicates that the GL theory well describes the out-of-plane anisotropy. Moreover, the sharp peak in the $H_{c2}(\theta)$ curve demonstrates that a very accurate field alignment is necessary to study intrinsic in-plane anisotropic properties since even a tiny out-of-plane misalignment can cause a sizable reduction of $H_{c2}$. In the present study, we emphasize that field misalignment is carefully avoided as shown in Supplementary Figs. 2, 3, and any extrinsic behavior originating field misalignment is absent (also see Method for quantitative discussion).

This high anisotropy is comparable to those observed in other typical Q2D superconductors such as cuprates, iron pnictides, organics, and ruthenates[46]. Thus, $CsV_3Sb_5$ is also classified as a Q2D superconductor. This is indeed expected from high out-of-plane resistivity anisotropy (~600)[26] and from cylindrical Fermi surfaces[27] reflecting the layered crystal structure. According to the GL theory, the anisotropy $H_{c2\|ab}/H_{c2\|c}$ equals to the anisotropy of the superconducting coherence length[3], and the latter is approximated by the anisotropy of the Fermi velocity. Thus, we reach the approximate relation $H_{c2\|ab}/H_{c2\|c} \sim \langle v_{F\|ab}^2 \rangle^{1/2}/\langle v_{F\|c}^2 \rangle^{1/2}$. Here, $\langle \cdots \rangle$ denotes the average over the Fermi surface. On the other hand, the resistivity anisotropy should be inversely proportional to the anisotropy in the average of the squared Fermi velocity, $\rho_{ab}/\rho_c \sim \langle v_{F\|c}^2 \rangle/\langle v_{F\|ab}^2 \rangle$, according to the standard transport theory. Thus, we obtain roughly $H_{c2\|ab}/H_{c2\|c} \sim (\rho_c/\rho_{ab})^{1/2}$. In $CsV_3Sb_5$, the out-of-plane transport anisotropy of 600 would naively lead to $H_{c2}$ anisotropy of 25, which is comparable to the observed $H_{c2}$ anisotropy of 10. Similar relation has

been reported in typical layered superconductors such as $YBa_2Cu_3O_{7-\delta}$ ($H_{c2\|ab}/H_{c2\|c} \sim 4$, $\rho_c/\rho_{ab} \sim 40$, $\sqrt{\rho_c/\rho_{ab}} \sim 6.3$)[47,48] and $Sr_2RuO_4$ ($H_{c2\|ab}/H_{c2\|c} \sim 20$–$60$, $\rho_c/\rho_{ab} \sim 2000$, $\sqrt{\rho_c/\rho_{ab}} \sim 45$)[49,50].

## In-plane hexagonal and nematic anisotropies

We now explain the novel in-plane anisotropy of $H_{c2}$, which cannot be accounted for by the ordinary GL formalisms. In Fig. 3a, we plot the in-plane field-angle $\phi$ dependence of $H_{c2}$ measured at 1.66 K. If the ERS is fully satisfied, no anisotropy would be visible. However, surprisingly, the $H_{c2}(\phi)$ data exhibit clear oscillation composed of six and two-fold components. Indeed, the data can be well fitted with the formula $H_{c2}(\phi) = H_{c2}^{(6)}\cos(6\phi) + H_{c2}^{(2)}\sin(2\phi) + H_{c2}^{(0)}$, as shown by the broken curve in Fig. 3a. Here the parameters $H_{c2}^{(i)}$ represent amplitudes of $i$-fold oscillatory components. The obtained parameters are $\mu_0 H_{c2}^{(6)} = 0.0107 \pm 0.0008$ T, $\mu_0 H_{c2}^{(2)} = 0.0079 \pm 0.0009$ T, and $\mu_0 H_{c2}^{(0)} = 2.0770 \pm 0.0006$ T. Figure 3b shows the relative sixfold component obtained by subtracting $H_{c2}^{(2)}\sin(2\phi) + H_{c2}^{(0)}$ from the data and normalized by $H_{c2}^{(0)}$. Similarly, the relative twofold component shown in Fig. 3c was obtained by subtracting $H_{c2}^{(6)}\sin(6\phi) + H_{c2}^{(0)}$ from the data. These data reveal that both anisotropy components are similar in size and of the order of 0.5% of $H_{c2}^{(0)}$. The raw $H_{c2}$ data, as well as the extracted components, are also shown as polar plots in Figs. 3d–f, in order to visually highlight the anisotropy behavior.

To evaluate the in-plane $H_{c2}$ anisotropy in a more effective and sensitive way, we measured the specific heat in the middle of the SC transition. We fixed the magnetic field in the middle of the SC transition in the $C(H)/T$ curve (as illustrated in Supplementary Fig. 5), and

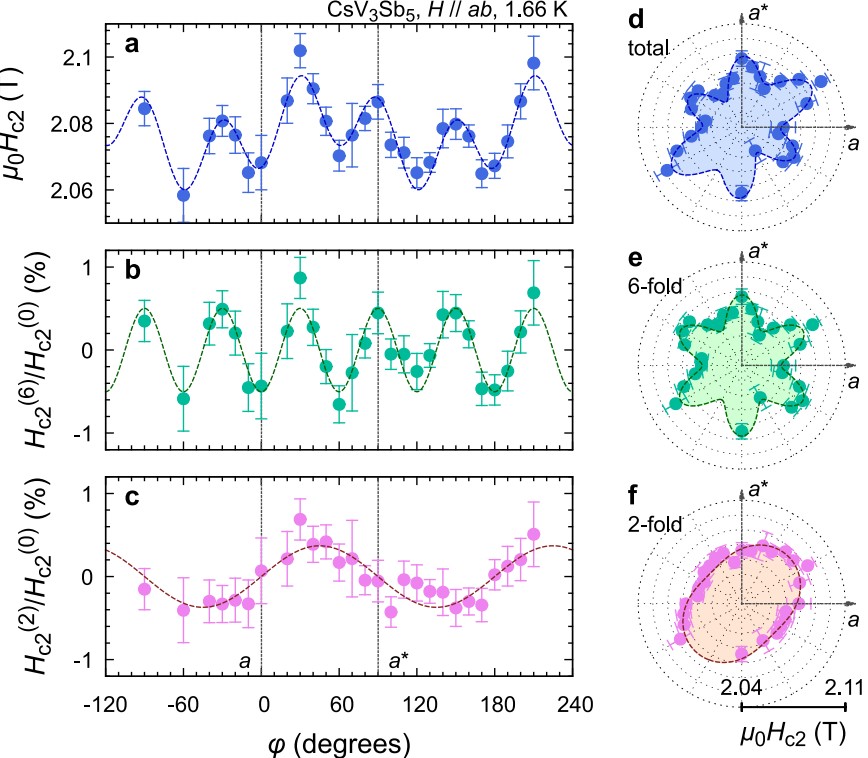

**Fig. 3 | Unconventional in-plane anisotropy of superconductivity in $CsV_3Sb_5$.**
**a** In-plane field-angle $\phi$ dependence of the upper critical field $H_{c2}$. The dotted curve shows the result of fitting using a combination of six- and twofold sinusoidal functions $H_{c2}(\phi) = H_{c2}^{(6)}\cos(6\phi) + H_{c2}^{(2)}\sin(2\phi) + H_{c2}^{(0)}$. The error bars represent asymptotic standard errors of $H_{c2}$ in the fitting process described in Supplementary Fig. 5. **b** Relative sixfold component obtained by subtracting $H_{c2}^{(2)}\sin(2\phi) + H_{c2}^{(0)}$

from the data and by normalizing with $H_{c2}^{(0)}$. **c** Relative twofold anisotropic component of $H_{c2}$ obtained by subtracting $H_{c2}^{(6)}\cos(6\phi) + H_{c2}^{(0)}$ from the data and subsequent normalization. **d**–**f** Polar plots of the total $H_{c2}$ (**d**), and its sixfold (**e**), and twofold components (**f**), illustrating the coexistence of sixfold anisotropy with superconducting nematicity. The broken curves with shadings inside illustrate the results of sinusoidal fittings.

then measured the field-angle dependence of the specific heat. If there is a small change in $H_{c2}$ as changing the field direction, we expect that $C(H)/T$ curves shift horizontally. Approximating that this shift is a parallel shift, we can estimate the $H_{c2}$ anisotropy from the measured specific heat anisotropy $\delta C/T$ using the formula $\delta H_{c2} \simeq -\alpha^{-1}(\delta C/T)$, where $\alpha = (1/T)(dC/dH)$ is the slope of $C/T$ vs $H$ curves at the SC transition. We also comment here that we have provided results of various control experiments, including $C(\phi)/T$ measurements above $H_{c2}$ (Supplementary Fig. 9), in order to exclude any extrinsic origins for the observed in-plane anisotropy. Details are discussed in the Method section.

In Fig. 4a–c, we plot the $H_{c2}$ anisotropy obtained by this method at representative temperatures. As one can see, the data set exhibits a combined six and two-fold oscillation, which is very similar to the data obtained from the direct $H_{c2}$ measurement shown in Fig. 3. Thus, these data were successfully fitted with the sinusoidal function $\delta H_{c2}(\phi) = H_{c2}^{(6)}\cos(6\phi) + H_{c2}^{(2)}\sin(2\phi)$ as shown with the solid curves, and the anisotropy terms $H_{c2}^{(6)}$ and $H_{c2}^{(2)}$ are extracted at each temperature. Data measured closer to $T_c$ are presented in Supplementary Fig. 9.

To completely confirm the absence of field misalignment, we performed $\theta$ sweeps at each $\phi$ as in Supplementary Fig. 2, and picked up the peak top value of each $C(\theta)/T$ curve, $C_{peak}/T$. This process means

that exact field alignment is checked each time after changing $\phi$ and $C_{peak}/T$ must be the specific heat when the field is exactly parallel to the $ab$ plane. In Supplementary Fig. 6, we plot $C_{peak}/T$ as a function of $\phi$. This each-point exact alignment data agrees quite well with the data obtained by the ordinary $\phi$ sweep data. From this agreement, we confirm that the observed in-plane anisotropy is intrinsic and free from any field misalignment effects.

The ratios $H_{c2}^{(6)}/H_{c2}^{(0)}$ and $H_{c2}^{(2)}/H_{c2}^{(0)}$ are rather important since they represent hexagonal and nematic SC anisotropies of the material, respectively. Thus, we plot the temperature dependence of these ratios in Fig. 5a, b. Data from direct $H_{c2}$ measurements (red points), from $C(\phi)$ measurements (blue points), and from the each-point exact alignment method (pink point), agree very well. It is quite intriguing that the hexagonal ratio $H_{c2}^{(6)}/H_{c2}^{(0)}$ stays almost constant up to 2.3 K, followed by a small downturn toward $T_c$ = 2.8 K. This downturn may be related to the small change in the slope of $H_{c2}(T)$ curve at around 2.3 K (Fig. 2c), which can be attributed to multi-band effects. The overall temperature dependence is in clear contrast to the standard GL theories predicting the ERS in hexagonal superconductors: As explained before, GL theories predict that $H_{c2}^{(6)}$ should be very small and, if exists, behave as $H_{c2}^{(6)} \propto (1 - T/T_c)^3$ as $T \to T_c$[15,16]. Since $H_{c2}^{(0)}$ in GL theories behaves as $H_{c2}^{(0)} \propto 1 - T/T_c$, the ratio $H_{c2}^{(6)}/H_{c2}^{(0)}$ should behave $\propto (1 - T/T_c)^2$ as shown with the broken curve in Fig. 5a. However, this prediction cannot explain the observed temperature dependence at all. We further examine the temperature dependence above 2.3 K, where the data exhibit downturn and approaches zero for $T \to T_c$. However, as shown in the log-log plot of $H_{c2}^{(6)}/H_{c2}^{(0)}$ vs. $1 - T/T_c$ (Supplementary Fig. 8a), the temperature dependence above 2.3 K is close

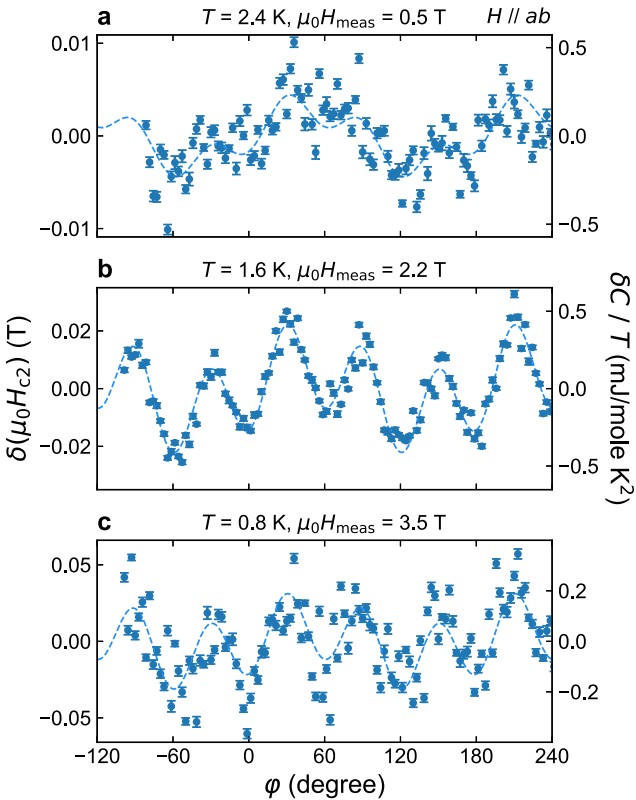

**Fig. 4 | Temperature evolution of in-plane superconducting anisotropy in CsV₃Sb₅.** Here, in-plane anisotropy of $H_{c2}$ deduced from the field-angle dependence of the specific heat at various temperatures are shown. The measurement field $H_{meas}$ is close to the mid-point of the SC transition at each temperature. **a** In-plane $H_{c2}$ anisotropy at 2.4 K. As explained in the text, the anisotropy $\delta H_{c2}$ is evaluated from the specific heat anisotropy $\delta C/T$ using the relation $\delta H_{c2} = -\alpha^{-1}\delta C/T$, where $\alpha = (1/T)(dC/dH)$ is the slope of the $C(H)/T$ vs. $H$ curve at the same temperature. The corresponding $\delta C/T$ values are shown using the right vertical axis. The curves represent the result of sinusoidal fitting using six and twofold oscillations. Each data point is obtained by averaging raw $C$ data for typically 120 s and the error bar represents standard errors of the averaged data set. **b** Same plot but for the data at 1.6 K. **c** Same plot but for the data at 0.8 K.

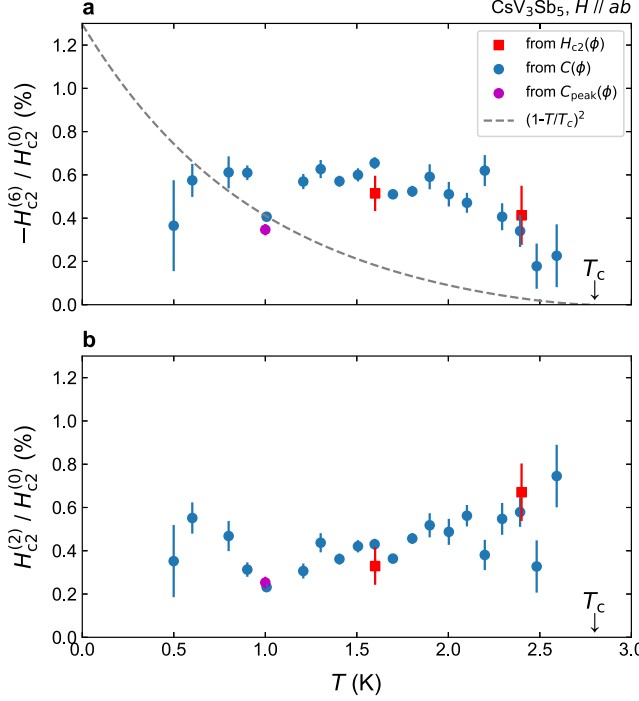

**Fig. 5 | Temperature dependence of in-plane hexagonal and nematic $H_{c2}$ anisotropy. a** Temperature dependence of the sixfold hexagonal anisotropy component $H_{c2}^{(6)}$ of $H_{c2}$ divided by $H_{c2}^{(0)}$, the averaged value of the in-plane $H_{c2}$. Note that −1 is multiplied so that the ratio becomes positive. The red squares are from $H_{c2}$-$\phi$ measurements as in Fig. 3, the blue circles are obtained from the $C$-$\phi$ measurements as in Fig. 4, and the purple circles are from the each-point exact alignment method shown in Supplementary Fig. 6. The error bar represents asymptotic standard errors of sinusoidal fittings of each $H_{c2}(\phi)$ data set. Clearly, the data deviate from the expectation under preserved ERS from the ordinary GL theory, $H_{c2}^{(6)}/H_{c2}^{(0)} \propto (1 - T/T_c)^2$ [13–15], which is plotted using a broken curve. **b** Temperature dependence of the twofold anisotropy component $H_{c2}^{(2)}$ divided by $H_{c2}^{(0)}$.

to linear and far from quadratic. This conclusion is further confirmed by examining the exponent $\alpha$ in $H_{c2}^{(6)}/H_{c2}^{(0)} \propto (1-T/T_c)^{\alpha}$ by performing fittings in various fitting ranges. As shown in Supplementary Fig. 8b, the exponent $\alpha$ does not reach 2, irrespective of the fitting range. Thus, this downturn is again hardly explained by the quadratic behavior predicted by the GL theory. This result, to our knowledge, provides the first thermodynamic evidence for ERS violation in hexagonal superconductors.

The nematic anisotropy ratio $H_{c2}^{(2)}/H_{c2}^{(0)}$ is also quite important and interesting. Firstly, this ratio remains finite even for thermodynamic measurements performed in the absence of field misalignment. This fact indicates that the nematicity in the SC state of $CsV_3Sb_5$ is intrinsic. Nevertheless, we should stress that this intrinsic in-plane anisotropy is as small as a fraction of a percent. This provides a clear demonstration that accurate field alignment is crucially important to evaluate intrinsic nematicity in materials with strong two-dimensionality. Secondly, the temperature dependence of $H_{c2}^{(2)}/H_{c2}^{(0)}$ does not reach zero for $T \rightarrow T_c$. This feature suggests that the SC nematicity is inherited from the nematicity already existing outside of the SC phase[51]. Indeed, various studies have revealed twofold rotational behavior in electronic properties in the CDW state of $CsV_3Sb_5$[27,32,33]. In that case, nematicity observed here is not a spontaneous feature of superconductivity.

## Discussion

Although nearly 30 years have passed since the ERS in hexagonal superconductors is theoretically pointed out, experimental observations of hexagonal $H_{c2}$ anisotropy, hinting at the ERS violation, have been quite limited[19,20,23]. Importantly, a detailed investigation of the temperature dependence of the hexagonal anisotropy is essential since conventional origins such as in-plane Fermi-velocity anisotropy can result in hexagonal $H_{c2}$ anisotropy that decays rapidly toward $T_c$ as $H_{c2}^{(6)} \propto [1-(T/T_c)]^3$. Moreover, previous experiments were performed using resistivity measurement, which can be affected by non-bulk effects such as surface superconductivity, as discussed in ref. 52. To the best of our knowledge, our study marks the first unambiguous observation of the violation of ERS in hexagonal superconductors using a bulk thermodynamic probe, which is free from extrinsic contributions such as surface or impurity effects.

Uncovering the origins of violation of the ERS, namely robust sixfold in-plane $H_{c2}$, in hexagonal superconductors, has been a big challenge in the past. So far, as summarized in Table 1, the only established explanation is a two-component SC parameter coupled with certain symmetry lowerings[14–17]. In the context of the hexagonal heavy Fermion superconductor $UPt_3$, where a two-component SC order parameter ($E_{2u}$ or $E_{1u}$ states) is believed to be realized[53,54], the sixfold $H_{c2}$ anisotropy observed by magnetotransport[20] has been attributed to the two-component superconductivity coupled to uniaxial symmetry breaking (weak antiferromagnetic order)[14,15], or to trigonal crystal distortion[16]. We comment that, for the former scenario, the direction of the uniaxial symmetry breaking is assumed to follow

the external field direction due to the magneto-anisotropy of the weak antiferromagnetism, and the sixfold SC anisotropy appears through coupling between superconductivity and antiferromagnetism, whose free energy changes by the field direction[14,15]. These theories predict violation of the ERS and the hexagonal $H_{c2}$ anisotropy to vary as $\propto 1-T/T_c$. A similar discussion has been developed more recently for doped $Bi_2Se_3$, where two-component nematic superconductivity is believed to be realized in a putative trigonal crystal lattice[17].

The present observation of sixfold $H_{c2}$ in $CsV_3Sb_5$ can be explained by a similar scenario, namely two-component superconductivity coupled with underlying orthorhombic and/or trigonal symmetry breakings. Theoretically, the possibility of two-component $d \pm id$ pairings have been discussed based on a model focusing on sublattice interference near the van-Hove singularities in the Kagome lattice[55,56]. A recent theory based on bond-order fluctuations also suggests the possibility of $p_x$ and $p_y$ wave pairings in the clean limit[29], which also belong to two-component order parameters. Moreover, in $CsV_3Sb_5$, the CDW order is known to exhibit electronic nematicity[27,32–34]. If such uniaxial electronic nematic order is rotatable by magnetic field direction, it can be one ingredient of the emergence of the clear sixfold $H_{c2}$ anisotropy, as the orthorhombic antiferromagnetism promotes the hexagonal $H_{c2}$ anisotropy in $UPt_3$[14,15]. As another possibility, if the lattice symmetry of the CDW phase is trigonal $P\bar{3}$ as discussed in ref. 28, such trigonal symmetry, together with a two-component SC order parameter, can also explain the hexagonal SC anisotropy as proposed in refs. 16,17. We should comment that a more recent investigation proposes that the global crystal structure in the CDW state is orthorhombic $Cmmm$[30]. To fully address the key symmetry reduction for the ERS violation, a thorough investigation of the relation between the subtle crystal structure and $H_{c2}$ anisotropy is required, which is left for future studies.

We should emphasize that this scenario of two-component superconductivity naturally explains the observed nematic anisotropy as well. When the two-component order parameter ($\eta_1$, $\eta_2$) is realized, the components $\eta_1$ and $\eta_2$ should be degenerate as long as the hexagonal or trigonal symmetry is preserved. In such cases, nematic anisotropy should be absent. However, under the presence of uniaxial anisotropy in the CDW state, as observed with many probes[32,33], the degeneracy between $\eta_1$ and $\eta_2$ must be lifted and can cause twofold nematic anisotropy in $H_{c2}(\phi)$. This is quite similar to the situation in the $Bi_2Se_3$-based nematic superconductors[8,10], where certain symmetry-breaking field is believed to cause large twofold anisotropy in SC quantities. In contrast, if an ordinary single-component SC state is realized, the twofold $H_{c2}$ anisotropy should occur through the anisotropy of the Fermi velocity $v_F$; i.e., $H_{c2\|a}/H_{c2\|a^*} = v_F^a/v_F^{a^*}$. The in-plane Fermi velocity anisotropy can be estimated from the in-plane field-angular dependent magnetoresistance (MR) measured with $c$-axis current flow[32]. Using a simple model with ellipsoidal Fermi surface[57], the angular MR $\rho_c(H)$ is related to the in-plane Fermi velocity as $v_F^a/v_F^{a^*} \sim \sqrt{[\rho_c^2(H \| a) - \rho_c^2(H=0)]/[\rho_c^2(H \| a^*) - \rho_c^2(H=0)]}$, as

**Table 1 | Violation of the emergent rotational symmetry in hexagonal superconductors**

| SC order parameter | Symmetry lowering | Emergent rotational symmetry | Refs. |
|---|---|---|---|
| Single-component | None | Preserved | 15,16,18 |
| Single-component | Orthorhombic | Preserved | 51 |
| Single-component | Trigonal | Preserved | 17 |
| Two-component | None | Preserved | 13–16 |
| Two-component | Orthorhombic* | Violated | 14,15 |
| Two-component | Trigonal | Violated | 16,17 |

Predictions based on GL theories are summarized for cases of single- and two-component SC order parameters with or without weak symmetry lowering from the perfect hexagonal lattice. When the emergent rotational symmetry is preserved, the hexagonal $H_{c2}$ anisotropy component, $H_{c2}^{(6)}$, should be small and behave as $\propto (1-T/T_c)^3$[13–15]. This principle holds under the existence of sixfold Fermi-velocity anisotropy[18]. In contrast, $H_{c2}^{(6)}$ should behave as $\propto (1-T/T_c)^1$ if the emergent rotational symmetry is violated[14–17], as observed in the present experiment in $CsV_3Sb_5$.
*Direction of the symmetry-breaking field must follow the magnetic field direction.

discussed in detail in Supplementary Note 1. From experimental data reported in ref. 32 under various temperature and field conditions, we obtained the in-plane Fermi-velocity anisotropy $v_F^a/v_F^{a'}$ of 9-14% (See Supplementary Table 1). Notice that this value may be overestimated since this model assumes an isotropic scattering rate and the anisotropy in MR is solely attributed to the Fermi-velocity anisotropy. In the CDW state, the actual scattering rate can be anisotropic if the charge ordering pattern has nematicity. Neglecting such possible contribution from anisotropic scattering rate, we would expect that the twofold $H_{c2}$ anisotropy $H_{c2}^{(2)}$ amounts 4–7% of $H_{c2}^{(0)}$. The observed nematic anisotropy of 0.3–0.8% is much smaller than this estimation. On the other hand, the twofold $H_{c2}$ anisotropy in this paper has the principal axis along $\phi = 45°$, whereas the angular MR and, thus, the Fermi velocity anisotropy exhibits principal axes along $\phi = 0$ or 90°. Such difference in the principal direction infer that the nematic $H_{c2}$ anisotropy originates from the SC order parameter, rather than from the Fermi-velocity anisotropy.

Nevertheless, we should comment here that some experiments support rather conventional SC order parameters[37–39]. If conventional pairing is truly realized, one needs to include other terms that are absent in previous GL theories[13–16], in order to explain the observed hexagonal anisotropy. For example, by taking into account the multiband nature of superconductivity, one may be able to explain hexagonal anisotropy. As another exotic possibility, roton pair-density wave (PDW), which has recently been proposed[42], may also explain the hexagonal anisotropy once its GL terms are included. Thus, we cannot deny novel coupling between conventional superconductivity and underlying electronic properties of the Kagome metal may explain the observed hexagonal anisotropy. But we would like to emphasize here that, regardless of the origin, the observation of ERS violation itself is very rare in any known hexagonal superconductors.

Our first thermodynamic evidence for violation of the emergent rotational symmetry together with nematicity clearly indicates that superconductivity in $CsV_3Sb_5$ is exotic. The most plausible explanation using existing GL theories is a two-component order parameter intertwined with CDW symmetry lowerings. The present work should stimulate further studies to search for possible unconventional SC properties originating from this unique order parameter, such as time-reversal-symmetry breaking and strong response to uniaxial deformation. The inclusion of novel ingredients, such as multi-band, CDW, and PDW contributions, may explain our observation even for single-component order parameters, which can be another direction of the theoretical frontier in the kagome metals $AV_3Sb_5$. Finally, our results shed light on the special importance of hexagonal superconductors, which can host unique states with the help of high rotational symmetry and intertwined electronic orders.

## Methods

### Sample preparation and characterization

High-quality single crystals of $CsV_3Sb_5$ were grown by a self-flux method with binary Cs-Sb as flux. The molar ratio of $Cs_{0.4}Sb_{0.6}:CsV_3Sb_5$ was chosen to be 20:1. The raw material was loaded in an alumina crucible, which was then sealed in an evacuated quartz tube with a vacuum of $2 \times 10^{-4}$ Pa. The tube was put into a muffle furnace and heated to 1000 °C at a rate of 5 °C/h. After maintaining at 1000 °C for 12 h, the tube was cooled to 200 °C at a rate of 3 °C/h and subsequently down to room temperature with the furnace switched off. The flux was removed by distilled water, and finally, shiny crystals with hexagonal shape were obtained.

After the growth, each crystal was examined by means of the Laue photographs and magnetization measurements. Laue photographs were taken by using a commercial Laue camera (Rigaku, RASCO-BL2) with a backscattering geometry, with incident X-ray along the $c$ axis. As exemplified in Fig. 1f, high-quality crystals exhibit clear Laue spots. The DC magnetization was measured using a commercial magnetometer

equipped with superconducting quantum interference device (SQUID) detection coils (Quantum Design, MPMS-XL). A sample was mounted in a plastic straw carefully so that we do not apply excessive strain that can degrade crystallinity. A crystal exhibiting sharp SC transition with nearly 100% volume fraction (Fig. 2a, b) was chosen and used in the following specific heat measurements.

### Calorimetry

The specific heat was measured using a homemade low-background calorimeter[8]. The sample was sandwiched by a heater and thermometer, for which commercial ruthenium-oxide resistance chips were used. The background contribution was measured using a silver plate as a reference sample and was subtracted from the data. We employed the AC method[58] to measure the heat capacity. The oscillatory component of the temperature $T_{AC}$ and its phase shift $\delta$ was measured by using lock-in amplifiers (Stanford Research Systems, SR830). Then the heat capacity $C$ is evaluated by the relation $C = [P_0/(2\omega_H T_{AC})]\sin\delta$[59], where $P_0$ is the power produced by the heater. We adjusted $P_0$ so that $T_{AC}$ is a few percent of the sample temperature. We examined the frequency of the AC heater current $\omega_H/2$ depending on field and temperature conditions to maximize the sensitivity and accuracy, since it is known that the optimal frequency depends on the sample heat capacity and on the thermal resistance between the sample holder and the thermal bath[58]. The temperature-sweep data was measured by changing the frequency in the range $\omega_H/2 = 0.2$–5.0 Hz at each temperature in order to check the absence of extrinsic frequency-dependent heat capacity. Subsequently, most of the field-strength and field-angle sweep data was measured with $\omega_H/2 = 0.2$ Hz, which is confirmed to be in the proper frequency range while we can apply sizable $T_{AC}$ without inducing too much temperature offsets. The calorimeter was placed in a $^3$He-$^4$He dilution refrigerator (Oxford Instruments, Kelvinox 25) and cooled down to around 0.2 K.

### Magnetic field control

The magnetic field was applied using the vector magnet system[45]. This system consists of two orthogonal SC magnets (vertical $z$ solenoid magnet up to 3 T and horizontal $x$ split magnet up to 5 T; Cryomagnetics, VSC-3050) placed on a horizontal rotating stage. By controlling the relative strengths of the currents to the $x$ and $z$ magnets, we can rotate the magnetic field vertically in the laboratory frame; and by rotating the stage, we can control the horizontal field direction.

The crystalline directions of the sample placed in the vector magnet was determined by making use of the $H_{c2}$ anisotropy. The $ab$ plane was determined by detecting the SC response in the specific heat while rotating the magnetic field in the laboratory frame with field strength close to the in-plane $H_{c2}$. As explained in Supplementary Figs. 2, 3, the accuracy between the determined and actual sample frames are better than 0.1° in the $\theta$ direction. We emphasize that the sharp peaks with the full-width half-maximum of around 1° in the $C(\theta)/T$ curves (Supplementary Fig. 2) allows such a very accurate field alignment. Then, within the determined $ab$ plane, the directions of the $a$ and $a^*$ axes were determined based on the sixfold $H_{c2}$ oscillation with the help of Laue photos taken before the cooling. The accuracy in the in-plane angle $\phi$ is around a few degree. After these processes, we can obtain a conversion matrix between the laboratory and sample frames. The field angles shown in this Article is all expressed in the sample frame.

As explained above, the out-of-plane field misalignment is less than 0.1° in $\theta$ direction. Assuming the anisotropic effective mass model determined from the data in Fig. 2d, misalignment of 0.1°, if exists, would results in extrinsic twofold $H_{c2}$ anisotropy of 0.0005 T at 1.16 K, which is as small as 0.016% of the in-plane $H_{c2}$ of 3.05 T. This is much smaller than the observed nematic anisotropy (0.2–0.6%; see Fig. 5b). Therefore, the extrinsic twofold anisotropy originating from field misalignment is totally negligible in our study. The absence of

misalignment is further confirmed by the comparison between the each-point exact alignment method and the ordinary $\phi$-sweep method (see text and Supplementary Fig. 6).

## Control experiments to exclude extrinsic origins

We have performed various control experiments to exclude any extrinsic origins for the observed in-plane anisotropy. Firstly, $C(\phi)/T$ in a field above $H_{c2}$ does not show any detectable anisotropy (Supplementary Fig. 9), excluding anisotropy originating from normal-state contributions. Also, the fact that $H_{c2}^{(6)}$ and $H_{c2}^{(2)}$ exhibits non-monotonic field-strength dependence across $H_{c2}$ manifests that the anisotropy is not attributable to the magnetoresistance of thermometers and heaters but is solely linked to the superconductivity of the $CsV_3Sb_5$ sample. Similarly, we observed that a tiny (-0.5°) field misalignment changes the $C(\phi)/T$ curve significantly (Supplementary Fig. 10). Such very high sensitivity to the field misalignment again excludes background or magnetoresistance contribution. We have also confirmed that in-plane anisotropy is not detectable if we use a reference sample (a silver plate), as shown in Supplementary Fig. 11.

## Data availability

The authors declare that the data used to draw figures in this paper and its supplementary information are available at the Figshare depository (https://doi.org/10.6084/m9.figshare.25288207). The other data that support the findings of this study are available from the corresponding author upon request. Source data are provided with this paper.

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

## Acknowledgements

We acknowledge Y. Yanase, G. Mattoni, S. Kitagawa, K. Ishida, J. W. F. Venderbos, D. Agterberg, and S. D. Wilson for fruitful discussion. The work at Kyoto Univ. was supported by Grant-in-Aids for Scientific Research on Innovative Areas "Quantum Liquid Crystals" (KAKENHI Grant Nos. JP20H05158 and JP22H04473: received by S. Yonezawa) from the Japan Society for the Promotion of Science (JSPS), Grant-in-Aids for Academic Transformation Area Research (A) "1000 Tesla Science" (KAKENHI Grant No. JP23H04861: received by S. Yonezawa) from JSPS, a Grant-in-Aid for JSPS Fellows (KAKENHI Grant No. JP20F20020: received by Y. Hu and S. Yonezawa) from JSPS, Grant-in-Aids for Scientific Research (KAKENHI Grant No. JP17H06136: received by Y. Maeno; KAKENHI Grant No. JP22H01168: received by Y. Maeno and S. Yonezawa) from JSPS, by Core-to-Core Program (No. JPJSCCA20170002: received by Y. Maeno) from JSPS, by Bilateral Joint Research Projects (No.JPJSBP 120223205: received by S. Yonezawa) From JSPS, by a research support funding from The Kyoto University Foundation (received by S. Yonezawa), by ISHIZUE 2020 and 2023 of Kyoto University Research Development Program (received by S. Yonezawa), and by Murata Science and Education Foundation (received by S. Yonezawa). S. Yonezawa acknowledges support for the construction of the calorimeter from Research Equipment Development Support Room of the Graduate School of Science, Kyoto University; and support for liquid helium supply from Low Temperature and Materials Sciences Division, Agency for Health, Safety and Environment, Kyoto University. The work at the Beijing Institute of Technology (BIT) was supported by the National Science Foundation of China (NSFC) (Grants Nos: 92065109: received by Z. Wang; 12321004, 12234003: received by Y. Yao), the National Key R&D Program of China (Grant Nos. 2020YFA0308800 and 2022YFA1403400: received by Z. Wang), the Beijing Natural Science Foundation (Grant No. Z210006: received by Z. Wang). Z. Wang thanks the Analysis & Testing Center at BIT for assistance in facility support.

## Author contributions

Z. Wang and S. Yonezawa designed this study. K. Fukushima, K. Obata, and S. Yonezawa performed specific heat measurements and analyses with the guidance of Y. Maeno. Z. Wang, Y. Li, and Y. Yao grew single crystalline samples. K. Fukushima, K. Obata, S. Yamane, and Y. Hu characterized samples with the guidance of Y. Maeno and S. Yonezawa. The manuscript was prepared mainly by S. Yonezawa and K. Fukushima based on discussion among all authors.

## Competing interests

The authors declare no competing interests.
