## [Peer Review File · Nature Communications]

Violation of Emergent Rotational Symmetry in the Hexagonal Kagome Superconductor CsV_3Sb_5REVIEWER COMMENTS

Reviewer #1 (Remarks to the Author):

In this manuscript, the authors describe a series of field and temperature-based magnetization and calorimetry experiments on single crystals of the kagome superconductor CsV₃Sb₅. They identify an emergent rotational symmetry which is decidedly non-trivial. Their findings support a potentially exotic superconducting state.

In general, this manuscript is well written, and the experimental setup is well thought out. The data seems robust and the analysis is reasonable. I find that this manuscript likely appropriate for publication in Nature Communications. However, many of the authors analysis is based on the fact that the structural distortion associated with the CDW is presumed to remain in the P-3 symmetry. 1) There are now multiple works; see <https://arxiv.org/pdf/2211.16602.pdf> or <https://www.nature.com/articles/s41567-022-01805-7> where the CDW symmetry breaking most likely causes a orthorhombic distortion in the underlying crystal structure. The initial work the authors are referencing was a minimal model based on the simplest possible distortion mechanism. However, I believe the distortion to orthorhombic will ultimately prove as the correct one. I think the authors need to consider modifying their manuscript to take into account these considerations – particularly to ensure the future robustness of their analysis.

2) Figure 3 and 4 need error bars. The guides to the eye are nice, but for readers to make their own conclusions, I think a more rigorous error analysis should be presented.

The most pressing concern is (1), and while I don't think it detracts from the novelty of the ERS violation, I think it needs to be addressed prior to publication.

Reviewer #2 (Remarks to the Author):

Review 41210_0

Fukushima et al. Have invested a significant experimental effort into the study of the angle-dependent variation of the upper critical field in the Hexagonal Kagome superconductor CsV₃Sb₅. Thanks to a very precise two-axis rotation stage the authors were able to measure heat capacity while continuously varying the sample orientation with respect to the external magnetic field. The overall presentation is very clean and well done.

The key observation in this work is a superposition of two- and six-fold oscillations in H_{c2} for field varied within the basal plane; interpret in terms of evidence for nematic and hexagonal characters of the superconductivity in this compound. The latter component the authors identify as emergent rotational symmetry. The appearance of a six-fold rotational variation in H_{c2} is associated with two-component pairing symm near T_c .

The findings provide experimental evidences for the unconventional nature or the order parameter in this compound. The identification of a multi- or at least two component order parameter originates from the finding that the T-dependence of $H(6)_{c2}/H(0)$ appears to be proportional to $1-T/T_c$. This contradicts predictions based on Ginzburg-Landau formalism. The overall effect is rather

subtle and therefore difficult to detect.

One of my main concerns here is that the reported violation of the predicted temperature dependence is not a strong enough point for the claims made. In Fig. 2 c,d the anisotropy between the inplane, and out-of-plane critical fields is reported. The overall ratio is of the order of 10 but exhibits a reduction with decreasing temperature. This reflects the two dimensionality of the compound. Even though the authors argue that variations in the Fermi velocity may be small, as there are no clear signs from ARPES reported I doubt that they are negligible. In particular, it would be interesting to see the variations in resistivity upon inplane rotation near T_c .

Next, in Extended Data Fig. 7 the authors compare the linear and quadratic T-dependencies with the data. The question here is for what T-range, or in other words how far away from T_c , the LG-theory stays valid and can provide a correct prediction. Considering only the first few (taking into account the large error bars) points I would not dare to distinguish between a linear or quadratic dependence. From my point of view it is also not clear how $H(6)_c2$ would evolve further away from T_c . Indeed, the authors mention that multiband contributions may have to be taken into account in order to get the theory right. Figure 2c provides experimental evidence for strong multiband effects: Namely, a clear opposite curvature in the $H_c2(T)$ curve close to T_c . Furthermore, the change of anisotropy $H_c2|_a/H_c2|_c$ with temperature may play a key role in the overall T-dependence observed in this study. Therefore, a straightforward interpretation of the overall T-dependence may be rather challenging.

I therefore cannot recommend a publication by Nature Communications at this point.

Further comments:

The authors state that: "...the temperature dependence of $H(2)_c2 / H(0)_c2$ does not reach zero for $T \rightarrow T_c$." In Fig. 5 The authors show multiple points representing the ratio with enhanced error bars. Maybe it would be more telling to also provide a supplementary figure including full $H(6)_c2$ components plotted against angle for temperatures approaching T_c . There is a strong deviation for the points recorded at 2.4 K; 2.5 K; and 2.6 K. From my point of view, it is difficult to state that the nematic component persists or subsides at T_c .

Why are there such strong differences in the error bars in Figure 5?

Further comments:

- Also include important references in the figure captions in order to link expectations to the experimental findings
- Make a decision in using the word data in plural/singular form and stay consistent.
- Write full sentences and make sure language has been checked.
- I highlighted a few places in the text where I feel the language could be improved.

Reviewer #3 (Remarks to the Author):

In the manuscript NCOMMS-23-04874-T the authors report on in plane anisotropy of the upper critical field in CsV_3Sb_5 , deduced from specific heat measurements.

The manuscript starts with some preliminary "characterization" results but those first paragraphs already contain a lot of vague or even incorrect statements such as :

- "This difference [taking about the T-dependence of the specific heat] infers highly anisotropic and multi-band superconducting gap structure..." : this statement is much too vague : what does

"multigap" here mean (two, three four gaps ?), why should the anisotropy modify the T-dependence of the specific heat ? Are there nodes or line nodes in the gap ? etc....

- "This high anisotropy indicates Q2D nature of superconductivity » : an anisotropy of 10 seems very far from infinity to me, so can we really talk about 2D ? Is this really consistent with the anisotropy of 600 observed in transport ? Why does the anisotropy decrease at low T ?...

- "This successful fitting of the anisotropic mass model indicates that Hc2 is dominated by the ordinary orbital pair-breaking effect » : this is incorrect, the angular dependence would be very similar even if Zeeman effects would play a role. The analytical form would indeed be (slightly) different but it would much probably be impossible to distinguish between the different scenarii.

The second part of the manuscript is then devoted to the in plane anisotropy of the upper critical. Both a 2-fold and a 6-fold anisotropy are reported.

As far as the 2-fold anisotropy is concerned, the main issue is - as stated by the authors - a possible extrinsic feature coming from a possible misalignment. The authors claim that they could reach an alignment better than 0.1° which looks very doubtful to me, given the noise/signal ratio which seems to be of the order of 1% (at best). The « fourth-order polynomial with even order terms » used to fit the data are not provided and no error bar is given on the « position of the maximum ». Neither do we know the error bars on the angle determination. Finally, the authors do not comment on a possible 2-fold anisotropy of the calibration of the thermometers themselves. Data in the normal state (at 0.9K) are provided but those measurements are neither performed at the same T nor at the same H and the author did not even provide data at Tc or slightly above Tc in Fig.5. To be more convincing, the authors should (i) provide data for the empty chip at the same T/H values and (ii) provide data with, let's say a 1° misalignment to show what would be the influence of such a misalignment. All conclusion drawn from this 2-fold component is then doubtful to me.

The 6-fold term is less controversial (on an experimental point of view). In this case, I do not see how the authors could link their measurements with the main statement : « we infer that this clear ERS violation with nematicity is best explained by multicomponent nematic superconducting order parameter in CsV3Sb5 intertwined with symmetry breakings caused by the underlying charge-density-wave order » ? At best they could say that their data indicate the presence of a 6-fold anisotropy up to Tc which does not fit with the theory of two-component superconductors in trigonal D3d crystals and that's it (even the expected temperature dependence in this case is not straightforward to me). As stated by the others other explanations are possible but how do the authors justify that « regardless of the origin, the observation of ERS violation itself is very rare » ?

Finally, I would like to comment on the « high resolution » specific heat measurements. For me, we are very far from « high resolution » data as a sensitivity better than 1/10000 can be reached in « high resolution » AC measurements and we are very far from this value here (at best 1/100). The information given on the calorimetry are just useless : what does « we typically chose frequency of the AC heater current $\omega H/2$ to be 0.1-0.5 Hz depending on the condition to maximize the sensitivity and accuracy » mean ?

Reply to Reviewer #1

[Comment 1-1]

In this manuscript, the authors describe a series of field and temperature-based magnetization and calorimetry experiments on single crystals of the kagome superconductor CsV₃Sb₅. They identify an emergent rotational symmetry which is decidedly non-trivial. Their findings support a potentially exotic superconducting state.

In general, this manuscript is well written, and the experimental setup is well thought out. The data seems robust and the analysis is reasonable. I find that this manuscript likely appropriate for publication in Nature Communications. However, many of the authors analysis is based on the fact that the structural distortion associated with the CDW is presumed to remain in the P-3 symmetry.

[Our reply 1-1]

We thank Reviewer #1 for his/her careful and constructive review of our manuscript. We are pleased that he/she highly evaluates our work by stating "In general, this manuscript is well written, and the experimental setup is well thought out. The data seems robust and the analysis is reasonable." and concludes that publication in Nature Communication is "likely appropriate".

For the final sentence, our actual claim is not limited to the P-3 symmetry crystal model, as we will describe below.

[Comment 1-2]

1) There are now multiple works; see <https://arxiv.org/pdf/2211.16602.pdf> or <https://www.nature.com/articles/s41567-022-01805-7> where the CDW symmetry breaking most likely causes a orthorhombic distortion in the underlying crystal structure. The initial work the authors are referencing was a minimal model based on the simplest possible distortion mechanism. However, I believe the distortion to orthorhombic will ultimately prove as the correct one. I think the authors need to consider modifying their manuscript to take into account these considerations -- particularly to ensure the future robustness of their analysis.

[Our reply 1-2]

We thank the Reviewer for this valuable comment. We agree that the crystal symmetry in the CDW state is important. On the other hand, as I understand, the actual crystal symmetry in the CDW state is a very subtle issue.

In the manuscript, we propose the trigonal P-3 crystalline distortion as one possible mechanism. However, the violation of the ERS should occur also in the orthorhombic symmetry reduction. As already discussed in the previous version of the manuscript, in UPT₃, orthorhombic magnetic order that promotes coupling between magnetic field and superconducting order parameter has been considered as a leading mechanism to induce hexagonal H_{c2} anisotropy. In CsV₃Sb₅, orthorhombic electronic nematic order may cause six-fold H_{c2} anisotropy through similar mechanism.

To fully clarify the key symmetry reduction, it is necessary to investigate the relation between the crystal structure and H_{c2} anisotropy, since the actual crystal structure can vary due to various extrinsic factors such as sample quality, cooling rate, or small strain due to a sample stage, etc. Such thorough investigation is beyond the scope of this manuscript. We improved the discussion on the key symmetry reduction for the ERS violation in the third paragraph of the Section "Origins of the Unconventional Anisotropies".

We would like to emphasize here that this manuscript reports the first thermodynamic evidence for the hexagonal H_{c2} anisotropy and thus has a large impact regardless of its detailed origin. We add a new paragraph in the beginning of "Origins of the Unconventional Anisotropies" Section in order to emphasize this point.

[Our action 1-2]

We improved the discussion on the key symmetry reduction for the ERS violation in the third paragraph of the Section "Origins of the Unconventional Anisotropies". We also add a new paragraph in the beginning of this Section in order to emphasize the importance of our discovery. We also add <https://arxiv.org/pdf/2211.16602.pdf> (Now Phys. Rev. Materials, 7, 024806) as Ref. [30] to the reference list; <https://www.nature.com/articles/s41567-022-01805-7> has been already cited as [34].

[Comment 1-3]

2) *Figure 3 and 4 need error bars. The guides to the eye are nice, but for readers to make their own conclusions, I think a more rigorous error analysis should be presented.*

[Our reply 1-3]

Following his/her suggestion, we re-analyzed the data and added error bars to Figs. 3 and 4. For Fig. 3, the error bar is defined as the asymptotic standard errors of H_{c2} during the fitting process described in Extended Data Fig. 5. For Fig. 4, the error bars represent standard errors of the averaged specific-heat data set (each data point is obtained by averaging raw specific heat data for typically 120 sec). For both data, the error bars are reasonably small and thus do not alter any of our conclusion.

I should also mention that the curves in these figures are not merely guides to the eyes but present results of sinusoidal fittings.

[Our action 1-3]

As attached below, we revised Figs. 3 and 4 to include error bars. We explain definition of the error bars in the captions of these figures.

Figures 3 and 4 in the revised manuscript.

[Comment 1-4]

The most pressing concern is (1), and while I don't think it detracts from the novelty of the ERS violation, I think it needs to be addressed prior to publication.

[Our reply 1-4]

We thank Reviewer #1 again for constructive suggestions that helped us to improve the manuscript. We now addressed all of his/her concerns and revised manuscript accordingly. We now believe that the manuscript is acceptable for publication.

Reply to Reviewer #2

[Comment 2-1]

Fukushima et al. Have invested a significant experimental effort into the study of the angle-dependent variation of the upper critical field in the Hexagonal Kagome superconductor CsV₃Sb₅. Thanks to a very precise two-axis rotation stage the authors were able to measure heat capacity while continuously varying the sample orientation with respect to the external magnetic field. The overall presentation is very clean and well done.

The key observation in this work is a superposition of two- and six-fold oscillations in H_{c2} for field varied within the basal plane; interpret in terms of evidence for nematic and hexagonal characters of the superconductivity in this compound. The latter component the authors identify as emergent rotational symmetry. The appearance of a six-fold rotational variation in H_{c2} is associated with two-component pairing symmenear T_c .

The findings provide experimental evidences for the unconventional nature or the order parameter in this compound. The identification of a multi- or at least two component order parameter originates from the finding that the T-dependence of $H(6)_{c2}/H(0)$ appears to be proportional to $1-T/T_c$. This contradicts predictions based on Ginzburg-Landau formalism. The overall effect is rather subtle and therefore difficult to detect.

[Our reply 2-1]

We thank Reviewer #2 for his/her critical and constructive review of our manuscript. We are pleased for his/her high evaluation of our work with statements such as "*The overall presentation is very clean and well done*".

Below, we provide point-by-point responses to the comments, and describe revisions associated with each comment.

[Comment 2-2]

One of my main concerns here is that the reported violation of the predicted temperature dependence is not a strong enough point for the claims made. In Fig. 2 c,d the anisotropy between the inplane, and out-of-plane critical fields is reported. The overall ratio is of the order of 10 but exhibits a reduction with decreasing temperature. This reflects the two dimensionality of the compound. Even though the authors argue that variations in the Fermi velocity may be small, as there are no clear signs from ARPES reported I doubt that they are negligible. In particular, it would be interesting to see the variations in resistivity upon inplane rotation near T_c .

[Our reply 2-2]

We thank the Reviewer #2 for the comment. We here presume that what he/she in the end requests is resistivity anisotropy upon in-plane field rotation near but above T_c , although he/she mention out-of-plane anisotropy in the middle of the comment.

Resistivity variation upon in-plane field rotation has been already reported by Xiang *et al.* for CsV₃Sb₅ [Nature Commun. 12 6727; Ref. 30 of our initial manuscript] with *c*-axis current flow. They observed two-fold oscillatory component in the field-angular-dependent magnetoresistance in the normal state below around 40 K.

The angular magnetoresistance (AMR) with *c*-axis current flow can be related to the in-plane Fermi velocity anisotropy. Sugawara *et al.* obtained analytical expressions of the AMR using a simple model with an ellipsoidal Fermi surface [Sugawara *et al.*, J. Phys. Soc. Jpn 76, 114706 (2007); added to the reference.]. According to this theory,

the z-axis resistance ρ_z under in-plane field rotation $\mathbf{H} = H(\cos\varphi, \sin\varphi, 0)$ obeys the relation $\rho_z(H, \varphi) = \rho_z(H=0)[1 + AH^2/v_F^2(\varphi + \pi/2)]^{1/2}$ with a field-independent coefficient A. Here, $v_F(\varphi + \pi/2)$ is the in-plane Fermi velocity perpendicular to the field direction. We assume ellipsoidal Fermi velocity anisotropy with the principal directions x and y. After some algebra, we obtain the relation

$$\frac{v_{F,x}}{v_{F,y}} = \sqrt{\frac{\rho_z^2(H \parallel x) - \rho_z^2(H = 0)}{\rho_z^2(H \parallel y) - \rho_z^2(H = 0)}}$$

By using this formula, we estimate the Fermi-velocity anisotropy in CsV₃Sb₅. As listed in the table below, AMR data by Xiang *et al.* reveals the Fermi velocity anisotropy $v_F^a / v_F^{a^*}$ of around 9-14%. Notice that this value may be overestimated since this model assumes isotropic scattering rate and the anisotropy in AMR is solely attributed to the Fermi-velocity anisotropy. In the actual CDW state, scattering rate should also be anisotropic if the charge ordering pattern has nematicity.

T	$\mu_0 H$	$\rho_c(H = 0)$	$\rho_c(H \parallel a)$	$\rho_c(H \parallel a^*)$	$v_F^a/v_F^{a^*}$ from Eq. (1)
2 K	4 T	119 $\mu\Omega\text{cm}$	241 $\mu\Omega\text{cm}$	226 $\mu\Omega\text{cm}$	1.09
2 K	6 T	119 $\mu\Omega\text{cm}$	322 $\mu\Omega\text{cm}$	289 $\mu\Omega\text{cm}$	1.14
10 K	5 T	134 $\mu\Omega\text{cm}$	292 $\mu\Omega\text{cm}$	264 $\mu\Omega\text{cm}$	1.14
15 K	5 T	151 $\mu\Omega\text{cm}$	300 $\mu\Omega\text{cm}$	278 $\mu\Omega\text{cm}$	1.11

According to the GL and BCS theories, the in-plane nematic H_{c2} anisotropy is related to the in-plane Fermi-velocity anisotropy as

$$\frac{H_{c2}(H \parallel a)}{H_{c2}(H \parallel a^*)} \sim \frac{v_F^a}{v_F^{a^*}}$$

Notice that, in our experiment, the in-plane two-fold H_{c2} anisotropy $H_{c2}^{(2)}$ is given by the half of the peak-to-peak anisotropy. Therefore, the 9-14% anisotropy in v_F listed in the table would correspond to the ratio $H_{c2}^{(2)}/H_{c2}^{(0)}$ of around 4.5-7%. This anisotropy is much greater than that observed in our experiment (less than 1%). Nevertheless, the magnetoresistance anisotropy has the principal axis along the a or a^* axis, whereas the principle axis of the H_{c2} anisotropy is 45-degree tilted from the a axis. Because of this qualitative difference, it is so far difficult to attribute the nematic H_{c2} anisotropy solely to the possible Fermi velocity anisotropy induced by the CDW order.

We should comment here on the six-fold H_{c2} anisotropy, which is the most important observation in our manuscript. In AMR, six-fold anisotropy is substantially smaller than the nematic anisotropy and is also strongly dependent on sample quality, which suggest strong influence of scattering rate and thus large uncertainty in the estimation. Moreover, as already explained in the manuscript, the six-fold H_{c2} anisotropy originates from sixth-order GL terms and thus results strong decay $H_6 \sim (1 - T/T_c)^3$ as $T \rightarrow T_c$. This principle holds for the H_{c2} anisotropy originating from the Fermi-velocity anisotropy (newly added reference [18]; V. H. Dao *et al.*, Euro. Phys. J. B 44, 183188.). Since, this temperature dependence contradicts with our observation, six-fold H_{c2} anisotropy cannot be attributed to the Fermi-velocity anisotropy.

[Our action 2-2]

To include discussion mentioned above, we added Supplementary Note 1 to explain estimation of the Fermi-

velocity anisotropy, including the table shown above. We also revised the 4th paragraph of Section "Origins of the Unconventional Anisotropies". To strengthen the argument, we add Ref. [18] (V. H. Dao *et al.*, Euro. Phys. J. B 44, 183188).

[Comment 2-3]

Next, in Extended Data Fig. 7 the authors compare the linear and quadratic T-dependencies with the data. The question here is for what T-range, or in other words how far away from T_c, the LG-theory stays valid and can provide a correct prediction. Considering only the first few (taking into account the large error bars) points I would not dare to distinguish between a linear or quadratic dependence. From my point of view it is also not clear how H_{c2} would evolve further away from T_c. Indeed, the authors mention that multiband contributions may have to be taken into account in order to get the theory right. Figure 2c provides experimental evidence for strong multiband effects: Namely, a clear opposite curvature in the H_{c2}(T) curve close to T_c. Furthermore, the change of anisotropy H_{c2} || a/H_{c2} || c with temperature may play a key role in the overall T-dependence observed in this study. Therefore, a straightforward interpretation of the overall T-dependence may be rather challenging.

I therefore cannot recommend a publication by Nature Communications at this point.

[Our reply 2-3]

As the Reviewer pointed out, error bars near $H = 0$ becomes inevitably large even with our precise field direction control system. This is because H_{c2} itself becomes rather small. But, still, in spite of relatively large error bars, it is difficult to fit the data with the conventional $H_{c2}^{(6)}/H_{c2}^{(0)} \sim (1 - T/T_c)^2$ dependence in Extended Data Fig. 8a (Extended Data Fig. 7 in the previous version). We also performed fitting with $(1 - T/T_c)^\alpha$ with allowing the exponent α to change during the fitting. Moreover, in order to avoid ambiguities due to fitting range, we add fit-range dependence of the exponent α in the panel **b** of Extended Data Fig. 8. This new graph infers that, irrespective of the fitting range, the exponent never reaches 2, the expectation from the standard GL theory.

We agree with the reviewer that the multi-band effect and small temperature variations needs to be taken into account for full understanding of the observed phenomenon. But such theoretical analysis requires complicated calculations taking into account the actual band structure, higher-order GL energies and so on. We believe that such theoretical consideration should be left for future studies by theory specialists.

We would like to emphasize here that this manuscript reports the first thermodynamic evidence for the hexagonal H_{c2} anisotropy and thus has a large impact regardless of its detailed origin. We add a new paragraph in the beginning of "Origins of the Unconventional Anisotropies" Section in order to emphasize this point.

[Our action 2-3]

We add a fitting result with $A(1 - T/T_c)^\alpha$ in Extended Data Fig. 8a as shown below. The fit-range dependence of the fitting parameter α is newly shown in the panel **b**. We also add explanations of these new analysis in the 4th paragraph of "In-plane Hexagonal and Nematic Anisotropies" Section.

Also, we added a new paragraph in the beginning of "Origins of the Unconventional Anisotropies" Section in order to explain that the ERS violation is quite rare and this work is the first thermodynamic proof of it in any reported superconductors.

Extended Data Figure 8 in the revised manuscript.

[Comment 2-4]

Further comments:

The authors state that: "the temperature dependence of $H(2)_{c2} / H(0)_{c2}$ does not reach zero for $T \rightarrow T_c$." In Fig. 5 The authors show multiple points representing the ratio with enhanced error bars. Maybe it would be more telling to also provide a supplementary figure including full $H(6)_{c2}$ components plotted against angle for temperatures approaching T_c . There is a strong deviation for the points recorded at 2.4 K; 2.5 K; and 2.6 K. From my point of view, it is difficult to state that the nematic component persists or subsides at T_c .

Why are there such strong differences in the error bars in Figure 5?

[Our reply 2-4]

The size of error bar depends on various factors and thus exhibits non-trivial temperature dependence. Near T_c , the error bar becomes larger because the H_{c2} itself becomes very small and its anisotropy $H_{c2}^{(6)}$ or $H_{c2}^{(2)}$ get even more small. For $T \ll T_c$, the anomaly in $C(H)/T$ at $H = H_{c2}$ becomes shallower and thus sensitivity of the specific-heat anisotropy to the H_{c2} anisotropy is reduced substantially. Therefore, the error bar tends to become larger both near T_c and $T \ll T_c$. In some data sets, error bars are larger due to unoptimized measurement conditions (such as the frequency and the magnitude of the heater current).

Following your request, we show δH_{c2} vs φ curve at temperatures between 2.4 K and 2.6 K. We admit that the data contains unavoidable scatterings, but both nematic and hexagonal components are still much above the fitting uncertainties.

[Our action 2-4]

We add Extended Data Fig. 7 to show the C/T vs φ data measured at 2.4, 2.5 and 2.6 K.

Extended Data Figure 7 in the revised manuscript.

[Comment 2-5]

Further comments:

-Also include important references in the figure captions in order to link expectations to the experimental findings

[Our reply 2-5]

Following this suggestion, we put references in figure and table captions.

[Our action 2-5]

We put reference [3] in the caption of Fig.1, [13,14,15] in the caption of Fig.5, and [13,14,15] [14,15,16,17] in the caption of Table 1.

[Comment 2-6]

-Make a decision in using the word data in plural/singular form and stay consistent.

[Our reply 2-6]

We thank the reviewer for reading the text carefully. Now we decided to use "data" as plural.

[Our action 2-6]

We modified the text so that "data" is plural (e.g. removing "s" from verbs).

[Comment 2-7]

-Write full sentences and make sure language has been checked.

-I highlighted a few places in the text where I feel the language could be improved.

[Our reply and action 2-7]

We thank the reviewer for detailed corrections. We modified the text following the suggestions.

Reply to Reviewer #3

[Comment 3-1]

In the manuscript NCOMMS-23-04874-T the authors report on in plane anisotropy of the upper critical field in CsV₃Sb₅, deduced from specific heat measurements.

The manuscript starts with some preliminary "characterization" results but those first paragraphs already contain a lot of vague or even incorrect statements such as :

- "This difference [taking about the T-dependence of the specific heat] infers highly anisotropic and multi-band superconducting gap structure..." : this statement is much too vague : what does "multigap" here mean (two, three four gaps ?), why should the anisotropy modify the T-dependence of the specific heat ? Are there nodes or line nodes in the gap ? etc....

[Our reply 3-1]

First of all, we thank the Reviewer #3 for reviewing our manuscript. Here we reply to the comments on the gap structure.

Firstly, we use "multigap" here to indicate that the gap magnitude and anisotropy varies depending on the Fermi surfaces. In CsV₃Sb₅, its Fermi surface can be roughly categorized into two groups: the cylindrical Fermi surface around the Γ point mainly originating from Sb p orbitals, and the surfaces with complicated structure located around the First-Brillouin zone boundary mainly originating from d orbitals of vanadium. Since the orbital character as well as the density of states of these Fermi surfaces differ significantly, it is expected that the gap character (amplitude, anisotropy, etc.) should be different. Such Fermi-surface dependent SC gap has been already found in various superconductors such as MgB₂, iron-based superconductors, Sr₂RuO₄, etc.

Secondly, the SC gap anisotropy strongly affect temperature dependence of bulk superconducting properties including specific heat, superfluid density etc. This is because thermally-excited quasiparticles can survive down to low temperatures if there are Fermi-surface portions with small SC gap. This results in power-law dependences of physical quantities, which are well established both experimentally and theoretically in the last few decades.

Nevertheless, the multi-gap nature or gap anisotropy in CsV₃Sb₅ has been already discussed in previous literatures (Ref. [35-39] of our initial manuscript) and is not the main subject of our manuscript. Thus, we decided to remove the "vague" sentence "*This difference....*" from the manuscript, and instead we just mention that this will be discussed elsewhere.

[Our action 3-1]

In the end of the Section "Calorimetry using a high-quality sample", we replaced the sentence "*This difference....*" with a new sentence "*These features contains information on the SC gap structure, which will be discussed elsewhere*".

[Comment 3-2]

- "This high anisotropy indicates Q2D nature of superconductivity"; : an anisotropy of 10 seems very far from infinity

to me, so can we really talk about 2D ? Is this really consistent with the anisotropy of 600 observed in transport ? Why does the anisotropy decrease at low T ?";

[Our reply 3-2]

For the first point, the observed anisotropy ratio $\Gamma = H_{c2//ab} / H_{c2//c}$ of 10 is of course far from infinity. However, this ratio is comparable to anisotropies observed in other well-established "quasi-two-dimensional" superconductors, such as layered cuprates: 4-20; organic superconductors: 5-20; Layered ruthenate Sr_2RuO_4 : 20-60 (depending on temperature) etc. (See Klemm's textbook: newly added Ref. [46]). For these materials, the term "quasi-two-dimensional" have been commonly used to refer to the situation that the out-of-plane anisotropy is larger than approximately 3-5, but still far from the exact 2D limit. Thus, we believe that our statement "*This high anisotropy indicates Q2D nature of superconductivity*" is appropriate.

For the second point, yes, the H_{c2} anisotropy is indeed closely related to the anisotropy of transport. According to the GL theory, Γ equals to the anisotropy of the superconducting coherence length: $\Gamma = H_{c2//ab} / H_{c2//c} = \xi_{ab} / \xi_c$. Then the latter is equal to the anisotropy of the Fermi velocity. Thus, we obtain an approximate relation $H_{c2//ab} / H_{c2//c} \sim \langle v_{F//ab}^2 \rangle^{1/2} / \langle v_{F//c}^2 \rangle^{1/2}$. Here, $\langle \dots \rangle$ denotes the average over the Fermi surface. On the other hand, the resistivity anisotropy should be inversely proportional to the anisotropy in the average of the squared Fermi velocity, $\rho_{ab} / \rho_c = \langle v_{F//c}^2 \rangle / \langle v_{F//ab}^2 \rangle$, according to the standard Bortzmann transport theory. Thus, to summarize, $\Gamma = H_{c2//ab} / H_{c2//c}$ should be roughly equal to $(\rho_c / \rho_{ab})^{1/2}$, i.e. the square root of transport anisotropy. In CsV_3Sb_5 , the transport anisotropy of 600 would naively lead to Γ of $\sqrt{600} = 25$. This is comparable to the observed Γ of 10. Discrepancies of factor 2-3 in this kind of simple analyses have been commonly reported in various layered superconductors. For example, in Sr_2RuO_4 , resistivity anisotropy is around 2000, whose square-root is 44, and Γ ranges from 20-60 depending on the temperature range (Newly added Refs. [49,50]). Similarly, $\text{YBa}_2\text{Cu}_3\text{O}_7$ shows Γ of 4 and resistivity anisotropy of 40 (Newly added Refs. [47,48]).

For the third point, the observed temperature variation in Γ is 23%. It has been known in various superconductors that Γ can vary depending on temperature, and such variation sometimes exceeds 100% (See Klemm's textbook, newly added Ref. [46]). The origins of such temperature dependent Γ include higher-order coupling effect, multi-band effect, Pauli effect, spin-orbit coupling, dimensional crossover etc. In the present case, we presume that the change is mainly due to the multi-band effect. Nevertheless, further detailed analyses/discussion are necessary to conclude it and should be left for future studies. Here, we just put one sentence stating that the observed Γ change is comparable to other various superconductors.

To address these points, we added one short paragraph in the end of the section "Quasi-two-dimensional superconductivity", with newly added references.

[Our action 3-2]

In the Section "Quasi-two-dimensional superconductivity", we removed the sentence "*This high anisotropy ...*" from the end of the 2nd paragraph. Instead, we added a paragraph in the end of this section to discuss the out-of-plane anisotropy in CsV_3Sb_5 in more detail. In this new paragraph, we added several new references.

In addition, we add one sentence stating that the observed Γ change is comparable to other various superconductors.

[Comment 3-3]

- *"This successful fitting of the anisotropic mass model indicates that H_{c2} is dominated by the ordinary orbital pair-breaking effect"; : this is incorrect, the angular dependence would be very similar even if Zeeman effects would play a role. The analytical form would indeed be (slightly) different but it would much probably be impossible to distinguish between the different scenarii.*

[Our reply 3-3]

The Reviewer #3 is correct that the polar angle dependence of H_{c2} in the presence of the Pauli effect would be difficult to distinguish from the purely orbital-limited case. We now removed the part " *H_{c2} is the ordinary orbital pair-breaking effect*" since this claim is not related to the main subject of this work. Although we believe that the dominance of the orbital depairing is correct judging from various other facts e.g. the shape of the $H_{c2}(T)$ curves, we leave detailed analysis of $H_{c2}(\vartheta)$ data for future publications.

[Our action 3-3]

We removed the statement " *H_{c2} is the ordinary orbital pair-breaking effect*".

[Comment 3-4]

The second part of the manuscript is then devoted to the in plane anisotropy of the upper critical. Both a 2-fold and a 6-fold anisotropy are reported.

As far as the 2-fold anisotropy is concerned, the main issue is - as stated by the authors - a possible extrinsic feature coming from a possible misalignment. The authors claim that they could reach an alignment better than 0.1° ; which looks very doubtful to me, given the noise/signal ratio which seems to be of the order of 1% (at best). The "fourth-order polynomial with even order terms"; used to fit the data are not provided and no error bar is given on the "position of the maximum". Neither do we know the error bars on the angle determination.

[Our reply 3-4]

We thank the Reviewer #3 for this important comment. The accuracy of the field alignment relies much on the sharpness of the anomaly in the data used to determine the field alignment. In the present case, the data in Extended Data Fig. 2 exhibits very narrow peaks with the full-width half maximum of 1 degree. By fitting these sharp peaks with the 4th-order polynomial $f(\vartheta) = A_0 + A_2(\vartheta - \vartheta_{\text{peak}})^2 + A_4(\vartheta - \vartheta_{\text{peak}})^4$, we determine field direction very accurately. Notice that this determination of ϑ_{peak} makes use of not only the peak top but also the "symmetry" of the peak. Fitted ϑ_{peak} has errors of typically ± 0.02 degrees.

To explain the field alignment more precisely as requested by the Reviewer, we modified the text as explained below.

[Our actions 3-4]

- (1) We explain the "fourth order polynomial" explicitly in the caption of Extended Data Fig.2.
- (2) Results of the "fourth order polynomial" fitting are shown with dotted curves in Extended Data Fig.2 as shown below. Also, resultant ϑ_{peak} are also shown with the triangles.

- (3) We added error bars in Extended Data Fig.3, as shown below.
- (4) We also added one sentence in "Magnetic field control" subsection (in Method) to explain the importance of the sharp peak in $C(\vartheta)$ to perform accurate alignment.

Extended Data Figures 2 and 3 in the revised manuscript.

[Comment 3-5]

Finally, the authors do not comment on a possible 2-fold anisotropy of the calibration of the thermometers themselves. Data in the normal state (at 0.9K) are provided but those measurements are neither performed at the same T nor at the same H and the author did not even provide data at T_c or slightly above T_c in Fig.5. To be more convincing, the authors should (i) provide data for the empty chip at the same T/H values and (ii) provide data with, let's say a 1° misalignment to show what would the influence of such a misalignment. All conclusion drawn from this 2-fold component is then doubtful to me.

[Our reply 3-5]

We agree with the Reviewer that angular magnetoresistance of the thermometer can result in extrinsic oscillation in the observed heat capacity.

On the request (i), we indeed performed reference runs before the actual measurements. We compare reference data with data with the CsV₃Sb₅ sample in Extended Data Fig. 11. (The reference run is measurement of an ordinary Ag foil. This is because our calorimeter is designed so that the thermometer and the heater sandwiches a sample to minimize the background contribution; we cannot perform the "blank" run without a sample.) In this figure, due to a limited number of available reference-data sets, we compare data measured with similar magnetic field strength but with different temperatures. Nevertheless, it is expected that extrinsic anisotropies originating from the calibration of the thermometer and heater should be larger at lower temperatures because magnetoresistance of Ru-oxide thermometer chips are known to become significant at low temperatures. Also, we should also comment that the apparent signal-to-noise ratio looks enhanced in the reference data because of large resistance of the thermometer at low temperatures, as well as the inverse relation between heat capacity and the actual measured quantity (i.e the temperature oscillation $T_{ac} (C \propto 1/T_{ac})$). Therefore, the data in the right panels provides the upper limit of the extrinsic anisotropy due to thermometer magnetoresistance.

As shown in the insets, the anisotropy in C of the reference data is absent with the resolution of 0.05 nJ/K², which is less than 2% of the observed oscillations with the CsV₃Sb₅ sample. At higher temperatures, the extrinsic anisotropy is expected to be smaller as explained above. Therefore, any spurious anisotropy due to thermometer calibration is negligible.

On the request (ii), we show in the figure below a comparison between φ dependence of the heat capacity with exact alignment and that with slight (~0.5deg) misalignment (Extended Data Fig. 10). The specific heat oscillation is quite sensitive to the misalignment. Such high sensitivity is not expected for extrinsic origins due to the AMR of the thermometer.

Also, as the Reviewer mentioned, we updated Extended Data Fig. 9 (Extended Data Fig. 8 in the previous version) to provide comparison of φ dependences in the SC state (3.2 T), at H_{c2} (3.4 T), and in the normal state (3.8 T) measured at the same temperature (0.9 K). Within this narrow field range, both the two and six-fold oscillation components exhibits non-monotonic field dependence; both are quite strong at $H = H_{c2}$, and disappears in the normal state. Such non-monotonic behavior again excludes possibility of background origins, and instead confirms that the oscillation originates solely from superconductivity.

In addition, in order to completely address the absence of field misalignment, we add a new data set (shown below and added as Extended Data Fig. 6). For this data set, we performed ϑ sweep at each φ , and then picked up the specific heat at the peak position. This means that accuracy of the field alignment is checked each time after we change φ . We found that the H_{c2} anisotropy evaluated from this new "each-point exact alignment" method (panel **a**) matches with the data obtained from ordinary φ -sweep method (panel **b**). This new data set proofs that the nematic anisotropy exists in the total absence of misalignment. We added this figure to Extended Data and added associated explanation to the main text. $H_{c2}^{(6)}$ and $H_{c2}^{(2)}$ obtained from this data set is added to Fig.5.

[Our actions 3-5]

- (1) On (i), we added Extended Data Fig. 11 to compare reference data with data with the CsV₃Sb₅ sample.
- (2) On (ii), we added Extended Data Fig. 10, which compares φ dependence of the heat capacity with exact

alignment and that with slight (~ 0.5 deg) misalignment.

- (3) We changed Extended Data Fig. 9 (Extended Data Fig. 8 in the previous version). Now it contains data measured slightly below H_{c2} , at H_{c2} , and slightly above H_{c2} , in order to show non-monotonic field dependence in the magnitudes of oscillations.
- (4) We added Extended Data Fig. 6 to show that the observed in-plane anisotropies are observed with "each-point exact alignment" method, manifesting intrinsic nature of our observation. We also add several sentences in the 3rd paragraph of "In-plane Hexagonal and Nematic Anisotropies" Section.
- (5) We add a subsection "Control experiments to exclude extrinsic origins" in Method, to discuss the control experiments shown in Extended Figs. 9-11. We also modified the last sentence of the 2nd paragraph of "In-plane Hexagonal and Nematic Anisotropies" Section.

All added/revised figures are shown below.

Extended Data Figures 6 and 9 in the revised manuscript.

Extended Data Figures 10 and 11 in the revised manuscript.

[Comment 3-6]

The 6-fold term is less controversial (on an experimental point of view). In this case, I do not see how the authors could link their measurements with the main statement : "we infer that this clear ERS violation with nematicity is best explained by multicomponent nematic superconducting order parameter in CsV3Sb5 intertwined with symmetry breakings caused by the underlying charge-density-wave order" ? At best they could say that their data indicate the presence of a 6-fold anisotropy up to T_c which does not fit with the theory of two-component superconductors in trigonal D3d crystals and that's it (even the expected temperature dependence in this case is not straightforward to me). As stated by the others other explanations are possible but how do the authors justify that "regardless of the origin, the observation of ERS violation itself is very rare"?

[Our reply 3-6]

As we already stated in the previous version of the manuscript, we do not exclude possibilities that the 6-fold H_{c2} anisotropy originate from a mechanism different from two-component superconductivity. But the ERS violation, temperature-robust 6-fold H_{c2} anisotropy in hexagonal superconductors, has not been observed unambiguously with bulk experiments in any superconductors although nearly 30 years have passed since the ERS is theoretically pointed out. Among hexagonal superconductors, UPt_3 , a leading candidate hosting two-component superconductivity, has been most thoroughly examined experimentally. But such experiments were performed using resistivity measurement and there are discussion that the data may be contaminated by non-bulk surface superconductivity (as discussed in newly added Ref. [50], N. Keller *et al.*, Phys. Rev. B **54**, 13188 (1996)). In contrast, calorimetry used in our study is fully bulk sensitive and free from complications due to surface effects. Thus, we believe that the establishing the violation of ERS with a careful bulk experiment is quite important, stimulating future theoretical and experimental studies.

To emphasize the importance of our experimental observation of ERS violation using a bulk thermodynamic probe, we added a new paragraph in the beginning of "Origins of the Unconventional Anisotropies" Section.

[Our action 3-6]

We added a new paragraph in the beginning of "Origins of the Unconventional Anisotropies" Section in order to explain that the ERS violation is quite rare and this work is the first thermodynamic proof of it in any reported superconductors.

[Comment 3-7]

Finally, I would like to comment on the "high resolution"; specific heat measurements. For me, we are very far from "high resolution" data as a sensitivity better than 1/10000 can be reached in "high resolution" AC measurements and we are very far from this value here (at best 1/100).

[Our reply 3-7]

Our resolution of the heat capacity is in the range 0.1-0.5% and this might be worse than most sophisticated AC calorimeters used in certain conditions. But our resolution is comparable or even better than typical low-temperature in-field calorimeters (e.g. Ref.[5] Sakakibara *et al.*, Rep. Prog. Phys. **79**, 094002 (2016)). Moreover, the

background is of the order of 1 nJ/K^2 at 1 K , which allows us to measure a very small sample. These are the reason why we initially used the word "high resolution" for our technique.

Nevertheless, since this is not related to the main topic of this paper, we decided to remove the word "high resolution".

[Our action 3-7]

We removed the word "high resolution" from the beginning of the 3rd paragraph of "Anisotropies in superconductors" Section.

[Comment 3-8]

The information given on the calorimetry are just useless : what does "we typically chose frequency of the AC heater current $\omega_H/2$ to be 0.1-0.5 Hz depending on the condition to maximize the sensitivity and accuracy" mean ?

[Our reply 3-8]

To be precise, we examined frequency of the AC heater current depending on field and temperature conditions to maximize the sensitivity and accuracy, since it is known that the optimal frequency depends on the sample heat capacity and on the thermal resistance between the sample holder and the thermal bath. The temperature sweep data was measured by changing the frequency in the range 0.2-5.0 Hz at each temperature in order to check the absence of extrinsic frequency-dependent heat capacity. Subsequently, most of the field-strength and field-angle dependence data shown in this paper was measured with 0.2 Hz, where the resultant heat capacity is confirmed to be frequency-independent while we can apply sizable T_{AC} without inducing too much temperature offsets.

Following his/her suggestion, we add such information in the main text.

[Our action 3-8]

We improved the explanation about the calorimetry in the "Calorimetry" subsection in Method.

List of additional changes

In addition to the changes explained above in “[Our action X-X]”, we made other revisions as listed below.

- The affiliations are changed due to movement of Shingo Yonezawa and Soichiro Yamane from Department of Physics to Department of Electronic Science and Engineering. (Not highlighted by color)
- Acknowledgement, including funding information, is updated.
- We add “Data Availability” statement after “Competing financial interests”.
- Reference list is updated (e.g. some arXiv papers are now published.) (Not highlighted by color)
- Minor changes to correct typos, to improve explanations etc.

REVIEWERS' COMMENTS

Reviewer #1 (Remarks to the Author):

At least with regards to my reviews, the authors have addressed concerns adequately. If their analysis holds in both the orthorhombic and hexagonal motifs, then I have no further issues.

Their efforts to address the other two reviewers also seem to evidence a concerted effort. If the fellow reviewers see the additional evidence as sufficient, I think the manuscript would be appropriate for publication.

Reviewer #2 (Remarks to the Author):

The authors have provided a substantial revision including additional data and information. They have answered my questions in full. The observed variations in the heat capacity with two-fold and 6-fold symmetry is indeed the first time revealed by a thermodynamic probe in a kagome superconductor. The detected T-dependence deviates from standard theories and calls for further investigations. The authors have carried out a very careful and comprehensive study of specific heat, and in their revised manuscript, they provide a rigorous analysis of potential systematic errors. I, therefore, recommend a consideration for publication in Nature Communications.

After hinting at the potential influence of the Fermi surface and, hence, the variation of Fermi velocities, the authors included previous transport results into their discussion in order to provide a rough estimation. Although, there seems to be a significant anisotropy expected from those estimations, the key point here is that the observed two-fold anisotropy exhibits a different orientation. Furthermore, the authors point out that the temperature dependence of the 6-fold anisotropy deviates significantly from what is observed by Dao et al in the now included reference [18]. Hence, it is suggestive of a distinct origin likely associated with the superconducting order parameter of the compound.

The improved explanations in the main text as well as in Supplementary Fig. 8 make things much clearer. My initial doubt has been removed by the more refined presentation of the T-dependence. I now also better see how the error bars can be explained. The revision improved the transparency of the results and claims.

In reading the responses to the report of referee #3, I can only repeat my initial assessment: This work is of very high quality and precision. The authors present their results in a clearly structured and very transparent way. The additional measurements with “each-point-exact-alignment”, for a slight artificial misalignment, and without a sample, shown in Supplementary Figs. 9-11, demonstrate the careful data collection. This removes doubts of experimental artifacts that may be responsible for the observed anisotropies.

The authors have improved readability and typos, too.

Reviewer #3 (Remarks to the Author):

The authors took into account my remarks in a quite satisfactory way (as well as those of the two referees as far as I can judge). In particular, the "control" experiments are now displayed in extended Fig. 10 & 11 and the unclear (or incorrect) statements have been removed from the text as requested. Even if the origin of the observed effects is not fully clear I totally agree with the authors stating that "regardless of the origin, the observation of ERS violation itself is very rare in any known hexagonal superconductors" and given the fact that superconductivity in the recently discovered Kagome compounds is a "hot topic" I would rather recommend this work for publication.